# Whole-Genome Identification of APX and CAT Gene Families in Cultivated and Wild Soybeans and Their Regulatory Function in Plant Development and Stress Response

**DOI:** 10.3390/antiox11081626

**Published:** 2022-08-22

**Authors:** Muqadas Aleem, Saba Aleem, Iram Sharif, Maida Aleem, Rahil Shahzad, Muhammad Imran Khan, Amina Batool, Gulam Sarwar, Jehanzeb Farooq, Azeem Iqbal, Basit Latief Jan, Prashant Kaushik, Xianzhong Feng, Javaid Akhter Bhat, Parvaiz Ahmad

**Affiliations:** 1Department of Plant Breeding and Genetics, University of Agriculture, Faisalabad 38040, Pakistan; 2Center for Advanced Studies in Agriculture and Food Security (CAS-AFS), University of Agriculture, Faisalabad 38040, Pakistan; 3Barani Agricultural Research Station, Fatehjang 43350, Pakistan; 4Cotton Research Station, Ayub Agricultural Research Institute, Faisalabad 38000, Pakistan; 5Department of Botany, University of Agriculture, Faisalabad 38040, Pakistan; 6Agricultural Biotechnology Research Institute, Ayub Agricultural Research Institute, Faisalabad 38000, Pakistan; 7Department of Clinical Pharmacy, College of Pharmacy, King Saud University, Riyadh 11451, Saudi Arabia; 8Independent Researcher, 46022 Valencia, Spain; 9Zhejiang Lab, Hangzhou 311121, China; 10Key Laboratory of Soybean Molecular Design Breeding, Northeast Institute of Geography and Agroecology, Chinese Academy of Sciences, Changchun 130012, China; 11International Genome Center, Jiangsu University, Zhenjiang 212013, China; 12Botany and Microbiology Department, College of Science, King Saud University, 8, Riyadh 11451, Saudi Arabia; 13Department of Botany, GDC, Pulwama 192301, India

**Keywords:** *G. max*, *G. soja*, ascorbate peroxidase, catalyze, legumes

## Abstract

Plants coevolved with their antioxidant defense systems, which detoxify and adjust levels of reactive oxygen species (ROS) under multiple plant stresses. We performed whole-genome identification of ascorbate peroxidase (APX) and catalase (CAT) families in cultivated and wild soybeans. In cultivated and wild soybean genomes, we identified 11 and 10 APX genes, respectively, whereas the numbers of identified CAT genes were four in each species. Comparative phylogenetic analysis revealed more homology among cultivated and wild soybeans relative to other legumes. Exon/intron structure, motif and synteny blocks are conserved in cultivated and wild species. According to the Ka/Ks value, purifying selection is a major force for evolution of these gene families in wild soybean; however, the APX gene family was evolved by both positive and purifying selection in cultivated soybean. Segmental duplication was a major factor involved in the expansion of APX and CAT genes. Expression patterns revealed that APX and CAT genes are differentially expressed across fourteen different soybean tissues under water deficit (WD), heat stress (HS) and combined drought plus heat stress (WD + HS). Altogether, the current study provides broad insights into these gene families in soybeans. Our results indicate that APX and CAT gene families modulate multiple stress response in soybeans.

## 1. Introduction

Being sessile in nature, environmental stresses pose a considerable threat to plants and are mostly unfavorable for their normal growth [1,2]. Drastic and harsh climate changes further intensify the severity of abiotic events, viz., water deficit, salt and toxic metal/metalloid stresses [3,4]. To combat these challenges, plants possess various physiological and metabolic defense mechanisms. Stress-related metabolic activities trigger generation of excessive reactive oxygen species (ROS) [5]. ROS impart important functions with respect to signaling to regulate stress response. However, excessive ROS production leads to modification of nucleic acids, lipids and proteins, which results in defective functioning of the cell and, eventually, plant death. Therefore, maintenance of optimal ROS levels is needed for proper cell signaling, which can be achieved through proper balance between ROS production and removal [6,7].

Under stress conditions, redox homeostasis in plants is maintained via the antioxidant system. The enzymatic system comprises ascorbate peroxidase (APX), superoxide dismutase (SOD), catalase (CAT) and glutathione-S-transferase (GST), whereas non-enzymatic compounds include reduced glutathione (GSH), ascorbic acid (AA), phenolics, α-tocopherol, carotenoids, flavonoids and proline [8,9,10]. APX is a highly active class of antioxidants and is a type I heme-carrying peroxidase involved in the ascorbate-glutathione pathway for the removal of excessive H_2_O_2_ in plants under normal and stress conditions [11]. The number of genes of the APX family varies depending on the species; 9 genes have been identified *in Arabidopsis thaliana* [12], 8 in *Oryza sativa* [13], 7 in *L. esculentum* [14] and 26 in *G. hirsutum* [15]. APXs are regulated by multiple gene families and are classified on the basis of their subcellular localization. APX family members have previously been reported in *Arabidopsis thaliana*, rice, tomato, sorghum and cotton based on their genome [13,14,15,16,17].

On the other hand, CAT is a redox enzyme containing the heme domain, and all aerobic species possess CAT [18]. CAT plays an important role in plant defense, development and senescence. The genes of this family catalyze the degradation of H_2_O_2_ into O_2_ and H_2_O, with a maximum turnover rate of 6 × 10^6^ molecules min^−1^ among other antioxidants in the cells exposed to environmental stress [18]. The hotspot site of H_2_O_2_ production is peroxisome; however, it is also produced in other subcellular structures, such as chloroplasts, mitochondria and cytosols of higher plants [19]. Catalases are classified into three groups based on sequence, quaternary structure, heme prosthetic group and subunit size: I, monofunctional catalases; II, catalase peroxidases; and III, nonheme catalases [20]. The CAT genes have been reported in various species, e.g., one in sweet potato [21] and castor bean [22]; two in barely [23]; three in tobacco [24], *Arabidopsis thaliana* [25], maize [26] and rice [27]; four in cucumber [28]; and seven in cotton [29]. In tobacco, a transgenic line overexpressing the *E. coli Cat E* gene showed higher irradiance tolerance under drought conditions. In *Arabidopsis thaliana*, *CAT1* and *CAT2* expression is enhanced under drought stress and, and CAT1 and CAT2 were observed as the main H_2_O_2_ scavengers under various stress conditions, in addition to playing essential roles in plant adaptation to these stress conditions [25]. In another study, an elevated amount of H_2_O_2_ was recorded in *CAT 2* knockout mutants of *Arabidopsis thaliana* in association with the development of necrotic lesions [30].

Soybean (*Glycine max* L.) is a key source of oil and protein (35–40% on dry matter base) for humans and animals. It is also used in biodiesel production and is an important source of lipids and secondary metabolites, which are used in various industries [31]. Soybean yield is severely threatened by multiple stresses, viz., drought, heat, salinity and other environmental stresses. For example, it has been reported that water deficit stress alone can reduce soybean yield by ~40%, and all growth stages of soybean are sensitive to this stress; this results in deterioration of seed quality [32]. Many researchers have confirmed that the combination of heat and drought stress is more devastating for soybean growth, yield and seed quality than the individual effect of each [33]. Wild soybean is considered an important source of genes used to develop stress-tolerant soybean cultivars. Thus, there is a dire need to explore wild and cultivated soybean for sustainable yields in a fluctuating environment. The sequenced genome of both wild and cultivated soybean has been available, making gene prediction tools and annotation publicly available. Multiple families such as WRKY, MADS-box, ERF, BURP, MYB and NAC have been studied in cultivated soybean to elucidate their role in countering biotic and abiotic stresses [34,35,36,37,38,39].

However, to the best of our knowledge, the *APX* and *CAT* gene families have not been characterized in wild or cultivated soybeans. Therefore, in this study, we attempted to systematically identify and compare the *APX* and *CAT* gene families in wild and cultivated soybeans by employing bioinformatics approaches. Physicochemical properties, evolutionary history, gene structure, protein motifs, chromosome location, gene structure, phylogenetic tree, conserved domains, cis elements, gene duplication and synteny analysis were discussed in detail. Apart from these physical properties of the APX and CAT gene families, their putative role under drought, heat and the combination thereof were explored in different plant tissues.

## 2. Materials and Methods

### 2.1. Database Searches, Sequence Retrieval and Analysis of Physical Properties 

The genome sequences, along with protein, gene sequences and annotation files, of selected species, viz., *Glycine max* (cultivar Zhonghuang 13) and *Glycine soja* (Genotype *W05*), were downloaded from the SoyBase database. We filtered sequence files according to the following criteria: (1) the longest transcript was selected to represent each locus, (2) coding sequences less than 150 bp were eliminated and (3) the genes encoding the incomplete domain and truncated protein were discarded [40]. Protein domain families of each protein were identified by the hidden Markov model (HMM) of Pfam [41]. Furthermore, the SMART database was used to identify the domain related to APX and CAT genes [42]. The amino acid sequences of *APX* and *CAT* were also used to estimate the molecular weight, protein length and isoelectric point with the Expasy server [43]. Subcellular localization of these genes was identified with the WegoLoc [44] and CELLO v.2.5 [45] online tools.

### 2.2. Classification of APX and CAT Genes of Soybeans and Paralogous Gene Pair Prediction

The amino acid sequences of all *APX* and *CAT* genes were aligned using the Clustal Omega tool, and the parameters were set as default. The resultant alignment was employed to analyze the evolutionary relationship of cultivated and wild soybeans through the neighbor-joining method using MEGA7 software [46]. The positions with missing data and gaps were deleted from the final dataset. Evolutionary distance was calculated with the Poisson correction method [47]. A bootstrap of 1000 was used for phylogenetic analysis. Genes in the form of pairs located at terminated nodes were marked as paralogous gene pairs, also supported by their respective bootstrap values.

### 2.3. Intron–Exon and Motif Analysis of APX and CAT Genes

The GSDS web tool was used for analysis of intron–exon structure via comparison of CDS and the corresponding genomic sequences of APX and CAT genes [48]. MEME tool was used to identify conserved motifs. By submitting protein sequences, a maximum of 10 motifs were identified, with a motif width range of 6–50 [49].

### 2.4. Promoter Analysis for Cis-Acting Elements

In both soybean species, 1000 bp upstream sequences from translation start sites of *APX* and *CAT* were downloaded. The cis elements were scanned from the promoter region of each gene using the PLANTCARE online tool [50].

### 2.5. Chromosomal Localization and Gene Duplication Analysis

Positional information of the *APX* and *CAT* genes in *G. max* and *G. soja* was obtained from the gene annotation file (GFF3). The position of each gene of the *APX* and *CAT* gene families on respective chromosome was marked using MapChart [51].

Gene duplication between *APX* and *CAT* genes was identified using the SDTv1.2sequence demarcation tool [52]. Sequences with more than 90% similarity were considered duplicated sequences. Tandem duplications were characterized as five or fewer genes departing from the homologous genes, whereas genes separated by five or more genes or scattered on different chromosomes are considered segmental duplication.

### 2.6. Determining Gene Duplication Event of Paralogous CAT and APX Genes

Multisequence pairwise alignment of duplicated *CAT* and *APX* genes was undertaken by Clustal Omega using the PAM weight matrix of MEGA7 [46]. The subsequent alignment was analyzed by DnaSP v5.10.01 software, and synonymous (Ks) and non-synonymous (Ka) analysis was performed to calculate Ka and Ks substitution rates [53]. The ratio of Ka/Ks was calculated by the SNAP web tool to explore codon selection activation during evolution. The probable time of the duplication event was measured using the formula T = Ks/2λ, with a value of λ = 6.1 × 10^−9^ for soybean [54].

### 2.7. Comparative Phylogenetic Analysis

For comparative analysis, *CAT* and *APX* genes from 11 legumes were compared with *G. max* and *G. soja CAT* and *APX* genes. Amino-acid-based multiple sequence alignment of *APX* and *CAT* gene families of 11 species was carried out by Clustal Omega (https://www.ebi.ac.uk/Tools/msa/clustalo/, accessed on 26 February 2021). *CAT* and *APX* genes from *Arabidopsis thaliana* were also incorporated for comparative studies. The neighbor-joining method was used to explore the phylogenetic relationship. The Poisson correction method was applied to estimate the evolutionary distances in the form of rate of amino acid substitutions per site [47]. To do comparative phylogenetic analysis of fourteen species, 86- protein sequences of APX genes while 28 protein sequences of CAT genes were used. Positions with >95% coverage were eliminated from the dataset, whereas >5% alignment gaps, ambiguous bases and missing data were entertained at any position. The APX dataset comprised a total 219 positions, whereas that of CAT was 485 positions. MEGA7 was employed for evolutionary analysis [46].

### 2.8. Comparative Analysis of Ortholog Gene Pairs Identified in Soybeans

The *APX* and *CAT* genes identified from cultivated soybean were compared with wild soybean *APX* and *CAT* genes. The latest cultivated soybean genome assembly (glyma.ZH13.), as well as the GFF3 file, was retrieved. Both cultivated and wild genome assemblies were employed to calculate synteny and collinearity and draw a dual synteny plot with the MCScanS program in TBTools (v1.0692).

### 2.9. Expression Pattern Analysis from Fourteen Soybean Tissues

Transcriptome data of paralogue gene pairs of *G. max* (Wm82.a1) with respect to *G. max* (ZH13.a1) extracted from a public soybean database were used to investigate the expression of *G. max APX* and *CAT* in various tissues [55]. The expressions of *APX* and *CAT* genes were examined on the basis of the RNA Seq-Atlas of 14 tissues (http://soybase.org/soyseq/, accessed on 27 September 2021). These RNA-seq data were previously generated and deposited in the SoyBase database by Shen et al. [55]. The experimental conditions used to generate these RNA-seq data are presented in detail by Shen et al. [55]. Data are presented in Appendix A. Genewise expression data were normalized to draw a HeatMap using the TBtools program.

For deeper insight into *APX* and *CAT* genes, we studied the response of soybean *APX* and *CAT* genes under water deficit (WD), heat stress (HS) and combined water deficit and heat stress (WD + HS). To this end, we analyzed the expression profiles of all soybean *APX* and *CAT* genes in response to WD, HS and combined water WD + HS stresses using publicly available RNA-sequence data related to these stresses. The genome-wide RNA-sequence data for WD, HS and WD + HS were downloaded from the Gene Expression Omnibus (GEO) database of the NCBI under accession number GSE153951 (from *G. max*) [56]. Data of paralogue and orthologue gene pairs of *G. max* (ZH13.a1) and *G. soja* (W05.a1) with *G. max* (Wm82.a4) were extracted using the gene symbol for the genome of *Wm82.a4.v1*. Then, expression data of these paralogue genes were used to compare the performance of *APX* and *CAT* genes under three stresses, viz., WD, HS and combined WD + HS.

### 2.10. Plant Materials, Stress Treatments and Tissue Sampling

Uniform and healthy seeds of wild (*Glycine soja*, genotype W05) and cultivated (Genotype-86-4) soybean were used to determine the drought stress response. These seeds were disinfected with 70% (*v*/*v*) ethanol and surface-sterilized for 3 h using sodium hypochlorite and hydrochloric acid (100 + 15 mL). The seeds of *G. soja* and *G. max* genotype were planted in a 1:1 (*w*/*w*) mixture of soil and vermiculite. Seeds were germinated in an incubator under controlled conditions by maintaining the daytime temperature at 25 °C, with a nighttime temperature of 23 °C, relative humidity of 60% and a 16/8 h photoperiod. After five days, the germinated seedlings were transferred to half-strength Hoagland solution. After two-weeks, seedlings were treated with 15% PEG-6000 [57]. Control and treated roots were harvested 4 h, 8 h and 12 h after treatment. A completely randomized block design (CRBD) was used to plant the soybean seedlings, and three biological replicates were used for stress treatment. All harvested root samples were frozen in liquid nitrogen and stored at −80 °C until RNA extraction.

### 2.11. RNA Extraction and Quantitative Real-Time PCR (RT-qPCR)

To determine the expression pattern of *APX* and *CAT* genes in wild and cultivated soybean, total RNA was extracted from the roots of soybean plants using the protocol suggested by the manufacturer of the RNAprep Pure Plant Kit (Tiangen, Beijing, China). Purity and concentration of the total RNA was determined by a Nanodrop ND-1000 spectrophotometer, and RNA integrity number (RIN) was measured using an Agilent 2100 bioanalyzer. cDNA was synthesized using a Prime Script™ RT reagent kit (TaKaRa, Kusatsu, Shiga, Japan) according to the manufacturer’s instructions. Quantitative real-time PCR (RT-qPCR) was performed for each cDNA template using AceQ qPCR SYBR Green Master Mix (Vazyme, Nanjing, China), following the standard protocol. PCR amplification conditions were set as follows: 95 °C for 3 min; 35 cycles of 95 °C for 30 s, 57 °C for 20 s and 72 °C for 20 s in a 20 µL reaction mixture. For each sample, three biological replicates were conducted. The polymerase chain reaction (PCR) results were normalized using the *Ct* value corresponding to the soybean actin gene *GsoActin-11* (*Glysoja.18G049932*) as an internal control. The relative expression level for each gene was calculated by the 2^−ΔΔCt^ [58]. All primers were designed using the gene script web tool (Appendix A).

## 3. Results

### 3.1. Identification of APX and CAT Subfamily Members in Soybeans

A systematic approach was used to explore the genes encoding each subfamily with publicly available datasets. To this end, the sequenced genomes of *G. max* and *G. soja* were used to detect genes encoding the *APX* and *CAT* subfamilies. Initially, 18 *APX* genes were detected in *G. max*, whereas 22 were detected in *G. soja*. After removing truncated sequences, 11 *APX* genes were identified in *G. max*, and 10 *APX* genes were identified in *G. soja* (Appendix A).

The nomenclature of these genes was set as *GmaAPX*/*GmaCAT*, with the decimals after chromosome number representing multiple genes on the respective chromosome. For example, *GmaAPX11.4* refers and *APX* gene located on Chr.11 in *Glycine max*, where 4 indicates fourth *APX* gene on Chr.11 [59]. The detailed information of the *APX* and *CAT* genes, including their NCBI gene ID, local gene ID, rename ID and protein domain family, as well as their description, chromosome number, location, orientation, protein size, molecular weight, PI and localization of the genes, is presented in Table 1. All the *APX* proteins carried a single domain, PF00041, and *CAT* proteins carried PF00199. The amino acid length of *G. max APX* proteins ranged from 232 (*GmaAPX11.2*) to 432 (*GmaAPX6.1*), with corresponding molecular weights varying from 26.64 (*GmaAPX11.2*) to 46.98 (*GmaAPX6.1*) kDa, respectively. Isoelectric points of *G. max APX* proteins ranged from 5.5 (*GmaAPX11.4*) to 9.08 (*GmaAPX11.1*). The amino acid length of *G. soja APX* proteins ranged from 250 AA (*GsoAPX11.4)* to 435 AA (*GsoAPX4.1*), and molecular weights ranged from 27.07 (*GmaAPX11.4*) to 47.32 (*GsoAPX4.1*) kDa, with an average weight of 33.96 kDa. Isoelectric points ranged from 5.51 (*GsoAPX11.4*) to 9.08 (*GsoAPX11.1*) (Table 1).

The amino acid length of *G. max* CAT proteins ranged from 434 AA (*GmaCAT6.1*) to 611 AA (*GmaCAT4.1*), with molecular weights varying from 49.79 to 69.88 kDa. Isoelectric points of detected *G. max* CAT proteins ranged from 6.26 (*GmaCAT6.1*) to 6.77 (*GmaCAT14.1 & GmaCAT17.1*). The length of amino acids of *G. soja* CAT proteins varied from 492 AA (GsoCAT6.1) to 494 AA (*GsoCAT17.1*), with corresponding molecular weights ranging from 56.74 to 57.38 kDa and isoelectric points varying from 6.77 to 6.93 (Table 1).

### 3.2. Phylogenetic Tree and Gene Structure Analysis of APX and CAT Proteins

An unrooted phylogenetic tree was drawn to reveal the evolutionary relationship of the *APX* and *CAT* family members in *G. max and G. soja.* The phylogenetic tree exhibited a distribution with *APXs* and *CATs* in distinct groups due to conserved domains within groups among the proteins (Figure 1A).

*APX* subfamily members were distributed into five distinct groups. In groups I, II and III, orthologue gene pairs were distributed on the same chromosome of the respective species, e.g., *GsoAPX11.3*/*GmaAPX11.3* and *GsoAPX12.1/GmaAPX12.1* in group I; *GsoAPX11.1/GmaAPX11.1, GsoAPX11.2/GsoAPX11.2* and *GsoAPX1.1/GmaAPX1.1* in group II; and *GsoAPX11.4/GmaAPX11.4* and *GsoAPX12.2/GmaAPX12.2* in group III. In group V, distinct genes, viz., *GsoAPX6.1* showed highly diverse behavior, exhibiting similarity with genes present on Chr.06 of *G. max* (*GmaAPX6.1*), whereas in group, IV, it showed more similarity with the genes of Chr.02 (*GmaAPX2.1*) and Chr.14 (*GmaAPX14.1*) of *G. max.*

CAT genes were distributed into three groups. As compared to *APXs*, in all the three groups orthologue gene pairs were distributed on the same chromosome of the respective species i.e., *GsoCAT4.1/GmaCAT4.1, GsoCAT6.1* and *GmaxCAT6.1* in group I, *GsoCAT17.1*/*GmaCAT17.1* in group II and *GsoCAT 14.1/GmaCAT14.1* in group III. The members of group I showed distinct behavior in homology, whereas paralogue gene pairs showed more similarity (*GsoCAT4.1/GsoCAT6.1* and *GmaCAT4.1/GmaxCAT6.1*) as compared to orthologue gene pairs (*GsoCAT4.,/GmaCAT4.1* and *GsoCAT6.1*/*GmaxCAT6.1*) (Figure 1A).

Gene structure of *APX* and *CAT* genes showed a similar pattern as the grouping in phylogenetic analysis. The number of introns of *APXs* varied in both cultivated and wild soybeans, ranging from 5 to 11 in cultivated soybean and from 7 to 11 in wild soybean. Furthermore, comparative gene structure analysis of cultivated and wild soybeans confirmed intron/exon conservation among APX genes of both species, with few exceptions. For instance, *GmaAPX4.1* has 11 introns, and its homolog, *GsoAPX4.1*, has 10 introns; similarly, *GsoAPX11.2* has five introns, whereas *GmaAPX11.2* has seven.

The gene structure of *CATs* in cultivated and wild soybean was also analyzed to explore the intron/exon number and position, providing evidence for expansion of the gene family and its evolutionary relationship with its ancestors (Figure 1B). Overall, all the *CAT* genes carried between four and nine introns, although the number varied between cultivated and wild soybean, i.e., four to nine in *G. max* and five to seven in *G. soja.* The maximum number of introns (nine) was observed in *GmaCAT4.1*, followed by seven in *GsoCAT4.1*. In *G. max*, a higher number of introns suggested that members of the *CAT* gene family gained additional introns during a polyploidization event (Figure 1B). The presence of larger numbers of introns in *GmaCAT* transcripts may enhance the recombination frequency and contribute to maintenance of the counterbalance of mutation bias. Along with number, variations in position and length of introns, CDS, and downstream and upstream position were also observed in orthologue pairs. These variations may lead to varying lengths of *APX* and *CAT* genes in both species (Figure 1B).

### 3.3. Conserved Domain Detection in APX and CAT Proteins

To identify conserved domains, analysis was performed via MEME tool by using the APX and CAT protein sequences. Among APX and CAT proteins, 10 diverse conserved domains were recognized. The presence of conserved domains in the same group probably indicates the related functions of proteins. In these domains, the amino acid length varied from 31 to 50. As shown in the phylogenetic tree, some of the APX groups contain four domains, whereas others possess few domains. The number of conserved domains among the five APX groups are, four (motifs 1, 3, 4 and 6) in groups I, III and IV; two to four motifs in group II; and one to two motifs in group V (Figure 1C). Furthermore, in all orthologue gene pairs of *G. max* and *G. soja*, the motifs were highly conserved, except *GmaAPX11.2/GsoAPX11.2.* All the members of the CAT gene family have highly conserved domains, except *GmaCAT6.1*, i.e., motifs 1, 2, 3, 5, 7, 8, 9 and 10 (Figure 1C). Furthermore, *GsoCAT6.1/GmaCAT6.1* showed different motifs, whereas as in the rest of the orthologues, the *CAT* motifs were almost same.

### 3.4. Chromosome Localization and Gene Duplication of APX and CAT Genes

The distribution of predicted *APXs* and *CATs* was virtually localized on *G. max* and *G. soja* chromosomes. The distribution of predicted genes was non-random across. In *G. max*, of twenty chromosomes pairs, *APX* genes were distributed on seven only chromosomes. Chr.01, Chr.02, Chr.04, Chr.06 and Chr.14 each carried one *APX* gene, whereas Chr.11 and Chr.12 possessed four and two genes, respectively. In *G. soja*, 10 APX genes were distributed on only 5 chromosomes only. Chr.01 and Chr.04 each carried one APX gene, Chr.06 and Chr.12 possessed two genes each and four genes were mapped on Chr.11. In *G. max* and *G. soja*, of twenty pairs of the chromosomes, CAT genes were mapped on only four chromosomes, viz., Chr.04, Chr.06, Chr.14 and Chr.17 (Figure 2).

In order to better understand the evolution of *CATs* and *APXs*, we further explored the gene duplication event via the SDTv1.2 sequence demarcation tool. The results showed two gene pairs of *CATs* originated via segmental gene duplications in *G. max (GmaCAT4.1*/*GmaCAT6.1* and *GmaCAT14.1*/*GmaCAT17.1)* and three in *G. soja* (*GsoCAT6.1*/*GsoCAT4.1*, *GsoCAT4.1*/*GsoCAT14.1* and *GsoCAT14.1*/*GsoCAT17.1*) (Table 2). For APX, four duplicated gene pairs were observed in *G. max*: *GmaAPX14.1*/*GmaAPX2.1*, *GmaAPX4.1*/*GmaAPX6.1*, *GmaAPX11.3*/*GmaAPX12.1* and *GmaAPX11.4*/*GmaAPX12.2*. In *G. soja*, three duplicated gene pairs were observed: *GsoAPX4.1/GsoAPX6.2 GsoAPX11.4/GsoAPX12.2* and *GsoAPX11.3/GsoAPX12.1* (Figure 2 & Table 2). No tandem duplication was found for either *APX* or *CAT* gene families. We also found that the expansion of the *APX* and *CAT* genes was occurred due to segmental duplication.

### 3.5. Estimation of Divergence Time of G. max and G. soja APX and CAT PGPs

The divergence time of *APX* and *CAT* PGPs was evaluated by calculating synonymous (Ks) and nonsynonymous (Ka) substitution rates per site per year. For *APX* PGPs, the Ka/Ks varies from 0.23 to 2.26, whereas for *CATs*, it ranges from 0.08 to 0.14. Ka/Ks values for all the segmental *CATs* gene pairs were <1, showing that the duplicated *G. max* and *G. soja* CAT genes evolved under purifying selection pressure following duplication events (Table 2).

The Ka/Ks value for *APX* duplicated gene pairs of *G. max*, except *GmaAPX14.1/GmaAPX2.1* and *GmaAPX4.1/GmaAPX6.1*, was <1, indicating that they evolved under the pressure of purifying selection. *GmaAPX11.3/GmaAPX12.1* and *GmaAPX11.4/GmaAPX12.2* PGPs of *G. max* were found to be involved in positive selection. The Ka/Ks value for all three *APX* duplicated gene pairs of *G. soja* was <1, indicating that purifying selection is the major evolutionary selection force for these genes (Table 2).

T = Ks/2λ was utilized to determine the approximate duplication time. Results indicated that paralogous genes of *APXs* are duplicated in *G. max* at 1.64 to 5.74 MYA, whereas in *G. soja*, duplication of *APX* PGPs occurred almost 7.38 MYA. In *G. max*, *CAT* genes originated recently—about 8.20 to 10.66 MYA—whereas in *G. soja*, with the exception of *GsoCAT4.1/GsoCAT14.1*, the remaining duplications were recent, i.e., 6.56 to 11.48 MYA. Duplication of *GsoCAT4.1/GsoCAT14.1* was found to have occurred almost 60.66 MYA, which is far from the duplication of other *CAT* genes of *G. soja*, as well as from that of *G. max* CAT PGPs (Table 2).

### 3.6. Cis Regulatory Elements

In the *APXs,* shortlisted elements were grouped into transcription-related elements, abiotic-stress-related elements, light-responsive elements, and hormone-related and biotic stress-related cis elements. *CAT* promoter sequences contained 58 types of cis elements, i.e., light-responsive elements, hormone-responsive elements, and biotic- and abiotic-stress-related elements: transcription, cell cycle, growth and development metabolism and food synthesis. Abiotic-stress-related elements carried anoxic-responsive (ARE), drought-responsive (as-1, MYC), water-responsive (Myb) and stress-responsive (STRE) elements; biotic stress carried wound- and pathogen-responsive elements (WUN-motif) and light-responsive elements, including Box4, WUN-motif and GT1-motif. ARE, Myb and STRE were more present in *G. max APXs* as compared to *G. soja* APX, whereas MYC was more present in *G. soja.* Similarly, ARE and MYC were more present in CATs of *G. max*, whereas MYB and STRE were more present in *G. soja CATs*. Other than this, transcription-related elements consisted of AT~TATA-box, CAAT-box and TATA-box, whereas phytohormone-responsive cis elements comprising ABA-responsive (ABRE) and ethylene-responsive (ERE) elements were also identified in promoter genes. Water-responsive (Myb), light-responsive (MYC) and transcription-related cis elements (CAAT-box and TATA-box) were found to be conserved in all the genes of *G. max* and *G. soja*. Phytohormone-responsive cis elements, viz., ABRE and ERE, associated with abscisic acid and ethylene-responsive elements (ERE) were found in most of the *APX* genes of *G. max* and *G. soja*. Putative cis elements of both *APX* and CAT gene families identified in cultivated and wild soybeans are listed in Figure 3. Furthermore, gene-wise details of cis elements are described in Appendix A.

### 3.7. APX and CAT Genes Comparative Relationship from G. max and G. soja with Other Legumes and Arabidopsis thaliana

Full-length protein sequences of 14 legumes, Glycine max, Glycine soja, Medicago truncatula, Lotus japonicus, Cicer arietinum, Pisum sativum, Arachis ipaensis, Arachis duranensis, Phaseolus vulgaris, Lupinus angustifolius, Vigna unguiculata, Vigna radiate, Vigna angularis and Cajanus cajan, along with model plant Arabidopsis thaliana, were used to generate a comparative phylogenetic tree. The peptide sequence of proteins is provided in Appendix A.

The phylogenetic tree of APX genes was divided into five major clades. Each clade is represented by a different color of branch lines in Figure 4. Among them, clade I is the largest group and contains proteins from almost all species. Phylogenetic analysis revealed that there was not equal representation of the APX proteins from the all the studied species within given clades. In clade I, thirty-five numbers belonging to thirteen species, except *L. Japonica*, were included. The highest number (five) of *G. max* and *G. soja* orthologue gene pairs was also present in clade I. Clade II comprises nineteen proteins, with representation of all species, except L. *Japonica*. Clade III comprises nine members, whereas while clade IV is the smallest clade, with eight members. Clade V includes 15 members. Tighter clustering was observed between *G. max* and *G. soja APX* genes than other species showing more similarity among the sequences of these species as compared to other legumes and model plant *A. thaliana* (Figure 4).

The phylogenetic tree of CAT genes was divided into three major clades: Clade I carries a total of ten members, including one protein from *C. ariatinum, M. trancatula, P. vulgaris, V. angularis, V. radiata* and *V. unguiculata*; and two from *G. max and G. soja*. Clade II possesses nine members, one each from *C. cajan, P. vulgaris, V. angularis, V. radiata* and *V. unguiculata*; and two from *G. max* and *G. soja*. Clade III comprises with six members, with three proteins from *A. thialana*, two from *A. duranensis* and one from *L. angustifolius* (Figure 5). None of functional CAT proteins were found in *L. Japonica* and *Pisum sativum*. In *L. Japonica*, four CAT protein sequences were found to be truncated; therefore, they were discarded in the phylogenetic analysis. In *P. sativum*, one CAT protein sequence was found to be truncated; therefore, it was discarded in the phylogenetic analysis (Appendix A). The CAT proteins of four species, viz., *P. vulgaris, V. angularis, V. radiata* and *V. unguiculata* grouped together in both clades, showing close association between these species. All the members in three clades were predicted to be localized in peroxisome.

### 3.8. Comparison of Copy Number of G. max APX and CAT Genes with G. soja

We conducted a dual-synteny, plot-based, in-depth comparative analysis of *APX and CAT* among the cultivated and wild soybeans genomes to determine the evolutionary mechanism. This comparison revealed that all the *G. max APXs* and *CATs* member have homologues in the *G. soja* genome. A total of 20 orthologues were identified for 11 *G. max APX* genes. For *CATs*, a total of 16 orthologue gene pairs were detected. A collinearity analysis of the *APX* and *CAT* gene pairs is presented in Appendix A. Most of the *APX* and *CAT* gene pairs detected among cultivated and wild soybeans are anchored to syntenic blocks that are highly conserved, spanning >300 genes (Figure 6).

### 3.9. Expression Pattern of APXs and CATs in different Underground and Aerial Tissues of G. max Science Identifiers

The RNA-seq data of fourteen soybean tissues were utilized to draw pattern of expression for *APX* and *CAT* genes. From the online database, nine *G. max APX* genes were detectable at the gene expression level among fourteen tissues (Appendix A), whereas two genes (*GmaAPX1.1* and *GmaAPX2.1*) shared similar paralogues with *GmaAPX11.2* and *GmaAPX14.1*. Most soybean *APX* genes have a broad spectrum of expression. The majority of *APX* genes exhibited variegated gene expression in all the studied tissues. Different genes exhibit abundant expression among various tissues, such as *GmaAPX11.4* in nodule and pod shell 14 DAF, and *GmaAPX12.2* in roots. Based on the expression data of nine paralogous pairs in fourteen soybean tissues, it was evident that expression diverges. For example, *GmaAPX11.4* was highly expressed in the nodule, whereas *GmaAPX14.1* was undetectable. The same expression pattern was observed in paralogous pairs; for example, both *GmaAPX11.4* and *GmaAPX12.2* showed similar expression in seeds at 10 DAF and 14 DAF. On the contrary, some of the APX genes, such as *GmAPX4.1*, *GmaAPX11.3* and *GmaAPX14.1* were expressed at a medium level in the developing pod tissues only, with almost negligible expression in other tissues. *G. max* CAT genes also showed diverged expression in all fourteen tissues (Figure 7). CAT genes, i.e., *GmaCAT17.1*, showed a maximum level of expression in root, nodule and flower tissues, with medium to low expression in pod and seed development stages. *GmaCAT6.1* and *GmaCAT4.1* showed medium levels of expression in developing pods, with negligible expression in other parts.

To gain more insights about the functioning of the *APX* and *CAT* genes of soybeans under WD, HS and WD + HS tolerance, all the *APX* and *CAT* gene expression profiles were reanalyzed under WD, HS and WD + HS stresses using publicly available RNA sequence data. Nine and four *APX* and *CAT* genes, respectively, were extracted from GSE15395. The expression of most *APX* and *CAT* genes are reduced or enhanced under these stresses (Figure 8). Heat stress caused upregulation of the *GmaAPX1.1* and *GmaCAT4.1* genes. In response to drought stress, *GmaAPX4.1, GmaAPX6.1*, *GmaAPX11.4* and *GmaAPX12.2* genes showed upregulation, whereas no CAT gene was upregulated. In under combined WD and HS conditions, the transcripts of *GmaAPX11.4, GmaAPX12.2, GmaCAT6.1, GmaCAT4* and *GmaCAT17.1* were significantly upregulated (Figure 8). The *GmaAPX1.1* gene was downregulated under combined WD + HS treatments. Data regarding all the studied genes is available in Appendix A.

### 3.10. Response of G. soja and G. max APX and CAT Genes to Drought Stress

In the present study, we randomly selected five genes of each *G. max* and *G. soja* APX from five groups, as well as three CAT genes of each *G. max* and *G. soja* (Figure 9), and verified their response toward drought stress treatments using RT-qPCR analysis. Under drought treatment, some of the genes were upregulated, and some were downregulated at some points during treatment. For example, the *GsoAPX1.1* and *GmaAPX2.1* genes were upregulated in response to drought stress at 4 h, 8 h and 12 h as compared to the control (0 h) (Figure 9). However, *GmaAPX11.3* was downregulated at different time intervals, whereas the expression of *GsoAPX11.3* was downregulated at 4h but upregulated at 12 h after stress. The expression of *GsoCAT4.1* was increased 8 h after stress, whereas the opposite trend was observed for *GmaCAT4.1* at the same time point. Moreover, CAT genes showed a differential expression in response to drought stress, whereas only *GmaxCAT17.1* showed a continuous upregulation at all treatment points (4 h, 8 h, 12 h) as compared to the control (0 h) (Figure 9).

## 4. Discussion

Before genome sequencing, *G. max and G. soja* were categorized as distinct species because *G. max* differs phenotypically from *G. soja* [31]. However, whole-genome sequencing revealed that these two species are close relatives at the genetic level; they possess only a single-nucleotide difference of 0.31% [60]. By considering the reasons for the deletion event, we determined that the genomic structural variations are relatively higher in *G. soja* (3.45%) [61]. Wild soybean possesses wide genetic diversity as compared to cultivated soybean.The genes and gene families that regulate seed composition traits, tolerance to abiotic stresses (drought and heat stress) and resistance to biotic stresses (disease and insect pest resistance) are abundant in *G. soja*. Therefore, wild soybean (an ancestor of *G. max*) is a valuable sources of beneficial traits for *G. max* [62]. Among the reactive oxygen species, H_2_O_2_ is mainly scavenged by the *APX* and *CAT* genes due to their high affinity toward H_2_O_2_, thereby protecting the cell from oxidative damage [63]. Therefore, comparative analysis of all *APX* and *CAT* genes present in cultivated and wild soybeans provides the basis for understanding and enhancing abiotic stress tolerance in cultivated soybean. Extensive genome sequencing paves the way to identify all *APX* and *CAT* genes at the whole-genome level. The aim of these studies was to determine the phylogenetic relationship within and among legume plant species, compare the gene structures, analyze the conserved motifs, identify cis elements in the promoter region, mark chromosomal locations, visualize gene duplication, calculate the divergence rate of duplicated genes and construct the syntenic relationship between cultivated and wild soybean [41].

In the current study, we identified the 11 and 10 APXs in cultivated and wild soybeans, respectively. Comparative studies of both species showed that some of the genes are present in cultivated soybean but absent in *G. soja* and the vice versa. For example, *APX* genes (*GmaAPX2.1 & GmaAPX14.1*) were identified in *G. max*, whereas their orthologues are missing in *G. soja*; similarly, on Chr.06 of *G. soja*, two *APX* genes were identified, but only one was identified in *G. max.* However, our result showed a greater number of *APX* genes than previous estimates in *G. max*, i.e., seven *APX* genes [41]. There are two major explanations for this disparity. First, the high-quality whole-genome sequence of soybean was assembled and entered into the soybean genome database. Secondly, the search methodologies used in the current study differ from those used in previous investigations. In this study, we employed the *glyma. Zh13.gnm1* genome to obtain the sequences from the SoyBase data source, whereas previous investigations used the BLAST-P tool to retrieve sequences. Hence, these two factors may have contributed to these disparities in the results between the two investigations. An equal number of *CAT* genes (four) were detected in cultivated and wild soybeans. Previous studies revealed fewer *CAT* genes in plants; for example, three *CAT* genes were reported in *Arabidopsis thaliana* [25] and four in cucumber [28]. We also reported four *CAT* genes in both species, which is equal to the number observed in cucumber [28].

The *APX* proteins of soybeans were distributed into five groups, which is in line with reports for other plant species, such as *Arabidopsis thaliana*, rice, maize, sorghum, tomato, cotton, brassica and resurrection species [5,41]. The phylogenetic tree of *CAT* genes classified them into three clades, in accordance with results reported by Hu et al. [63], who also found *CAT* genes in cucumber classified into three groups by phylogenetic analysis. Furthermore, *GmaxCAT4.1*/*GsoCAT4.1*, *GmaxCAT6.1*/*GsoCAT6.1, GmaxCAT7.1*/*GsoCAT7.1* and *GmaxCAT14.1*/*GsoCAT14.1* showed close associations with one another (Figure 4). Phylogenetic tree analysis showed that *APX* and *CAT* genes of cultivated and wild soybeans are near to each other in every clade relative to other species, verifying the high similarity among the cultivated and wild soybeans, as previously observed by Aleem et al. [57]. In the *APX* and *CAT* gene families, the characteristics of gene structure (exon/intron) were relatively conserved within the phylogenetic group members of cultivated and wild soybean, with few exceptions; these findings are similar to those reported for other gene families in soybean, such as WRKY [41,64], F-box [65] and vascular plant one-zinc finger [66]. Overall variations were observed in terms of number, length and position of introns, as well as CDS of both *APX* and *CAT* genes in wild and cultivated soybean. These variations may lead to different lengths of *APXs* and *CATs* genes in both species. Furthermore, the higher number of introns in *G. max* relative to *G. soja* suggests that *CAT* genes gain additional introns during polyploidization events; the increased number of introns in stress-related genes increases the time required for their transcription, which in turn delays the stress response [67]. Regarding motif compositions of APXs and *CATs* of both species, little variation was observed. A similar conserved domain was observed among members of *G. max* and *G. soja* present in the same phylogenetic group. This signifies that the evolution of *G. max* takes place from *G. soja.* For 11 *APX* genes of *G. max,* 20 orthologue pairs were identified in *G. soja*, whereas for the *CAT* gene family, 16 orthologue gene pairs were identified. Similar findings were reported in other crops [68]. These findings suggest a close relationship between these two species, and a higher level of sequence similarity depicts that although cultivated soybean differs considerably from wild-type soybean, at the genome level, they are quite similar, with sequences found to be conserved from wild to cultivated species. Schmutz, et al. [69] also stated that the wild progenitor (*G. soja*) is a close relative and originator of *G. max*, sharing 97.65% genome similarity.

During evolution, duplications of genes are major factors resulting in species differences in genes. These differences become more obvious when the species are subjected to selection pressure and restrictive growth conditions [70]. Our study showed that no tandem, proximal or dispersed duplications were observed in the *CAT* and *APX* gene families of either cultivated or wild soybean. Segmental duplication was identified as a major driving force of duplication for these gene families in both cultivated and wild soybean. Previous studies reported segmental duplication as a major factor responsible for expansion of the *CAT* gene family in other species, i.e., cotton [29]. These duplicated genes may contribute to improved stress tolerance in plants [71]. Most duplicated genes have been silenced for many years; few survive and undergo purifying selection after duplication [55]. Positive selection leads to the accumulation of favorable mutations and dispersion in the population; on other hand, negative selection removes deleterious mutations [72]. In the two studied species, different selection forces operated during evolution of duplicated gene pairs of *APX*, but for CAT, only purifying selection operated in both species. In *G. max*, purifying and positive selection forces operated during duplication events of *APXs.* Purifying or negative selection has emerged as a primary driving force for duplication of *APXs* in *G. soja* [15]. Previous studies reported that gene duplication can relax selection and accelerated evolution as if mutation occurs in one of two duplicated genes and degrade its phenotypic function, it will not impact the reproduction of the organism so long as the other copy remains intact. Further, natural selection will not able to distinguish that which paralog should be under selection and which one free from selective constraints [73].

Furthermore, the divergence time of the duplication event also varied not only in APX duplication events but also in CATs. A higher value of Ka/Ks is associated with high divergence in the duplication event [74]. The segmental duplication events of *G. max APX* ranged from 1.64 to 5.74 Mya, and in *G. soja*, APX gene duplication events occurred ~7.38 million years ago. For *G. max CAT* genes, duplication events ranged from 8.20 to 10.66 Mya, whereas for *G. soja CAT* genes, it ranged from 6.56 to 60.66 Mya (Table 2). These results show that the duplication in *G. max* occurred more recently as compared to *G. soja* for both gene families. These findings are nearly equal with results reported by Lynch [54], who suggested that duplication in the glycine lineage occurred 13 million years ago.

As efficient ROS scavengers, APX and CAT proteins work against multiple biotic and abiotic stresses. Previous studies reported that *APX* and *CAT* genes play regulatory roles with respect to tolerance against multiple stresses, such as salt, heat, drought, UV radiation and oxidative stress [15,29,75]. We screened the upstream region of *APX* and *CAT* genes of both species to mark the cis elements. In the *CAT* gene family, besides the growth-related elements, biotic- (wound and pathogen) and abiotic-stress (ARE, as-1, MYC, Myb, STRE, Box4, WUN-motif and GT1-motif)-responsive cis elements were identified in both species. Similarly, in the *APX* gene family, cis elements, viz., ARE, as-1, MYC, Myb, STRE, Box4, WUN-motif, ERE, ABRE and GT1-motif related to biotic and abiotic stress responses were also identified. Almost all classes of cis elements were identified in both species, although overall, they vary in terms of their number. Some of the cis elements were found to be more present in *G. max*, whereas others were more present in *G. soja*, which indicates a response of both these species against multiple biotic and abiotic stresses. Regarding the functioning of these APX and CAT families, various studies in plants have reported that *CAT* genes are significantly induced by drought, salinity, cold and ABA treatments [61,76], whereas APX has been reported to work against drought, salinity, cold, ABA treatment, high light, temperature fluctuation and metal toxicity [76].

To further verify the expression pattern of *APXs* and *CATs* in response to plant stress conditions, we studied the expression pattern of the APX and CAT genes under stress conditions using RNA-seq data (previously published) and RT-qPCR analysis. The RNA-seq-based expression pattern at different stages of soybean growth and tissues in *G. max* showed tissue-specific expression of *APXs* and *CAT* genes. For example, *GmaAPX11.4* and *GmaAPX12.2* showed enhanced expression in all tissues (above-ground and underground tissues), along with high expression levels at various stages of pod and seed development (Figure 6). Our results show demonstrate the functional distribution of different members of the *APX* and *CAT* families in response to various tissue/developmental stimuli, as previously documented by Caverzan et al. [77]. Previous findings showed that *APX* and *CAT* genes are crucial for plant responses to various stresses, such as drought, heat and salt, as reviewed by Sofo et al. [76]. In this study, expression profiles of the *APX* and *CAT* genes under WD, HS and WD + HS using the published data of Shen et al. [55] indicated that soybean *APX* and *CAT* genes function extensively in responding to WD and HS. Stress-specific responses were also reported for these genes; for example, *APX* genes, viz., *GmaAPX4.1, GmaAPX6.1, GmaAPX11.4* and *GmaAPX12.2*, were upregulated under WD, whereas *GmaAPX1.1* and *GmaAPX11.1* were responsive under HS; a similar case was observed for *CAT* genes. This indicates that different *APX* and *CAT* genes function to regulate drought and heat stress responses, as previously reviewed and documented in different plant species by Caverzan et al. [77].

We also conducted a wet-lab experiment and RT-qPCR analysis to study the expression patterns of *APX* and *CAT* genes in response to drought stress in both cultivated and wild soybean. In the present study, we randomly selected five homologue genes of *GmaAPX* and *GsoAPX* from cultivated and wild soybean, respectively; our results showed significant differential responses of the respective homologous genes of cultivated and wild soybeans under drought treatments. We also selected the four homologous genes each of *GmCAT* and *GsoCAT* in cultivated and wild soybean, respectively; the respective homologous genes showed differential expression under drought stress in cultivated and wild soybeans. These results indicate that differential expression of the *APX* and *CAT* genes in response to stress conditions among the cultivated and wild soybeans might be responsible for the enhanced resistance of wild soybean against plant stress conditions, as previously reported by Aleem et al. [78]. APX and CAT genes are important components of the plant antioxidant defense system, and their increased gene expression assists in the scavenging of the excess ROS [76], preventing oxidative damage of plant tissues resulting from the various abiotic stresses [79].

## 5. Conclusions

The current study is the first comprehensive and systematic report of genome-wide characterization of *APX* and *CAT* gene families in cultivated and wild soybeans. A comparative study of *APX* and *CAT* genes in the cultivated and wild soybeans has not been conducted previously. This study provides a basic genomic resource with respect to the antioxidant genes, viz., *APX* and *CAT*, that participate in ROS scavenging under oxidative stress produced by various biotic/abiotic plant stresses, preventing oxidative damage to plants. This study provides future avenues to elucidate the molecular mechanism underlying *APX*- and *CAT*-mediated stress tolerance in soybean. Proper gene function validation of these identified *APX* and *CAT* genes under multiple biotic and abiotic stresses using the overexpression or CRISPR techniques will allow for their effective utilization in the development of stress-tolerant varieties of soybean. Finally, our comparison of the *APX* and *CAT* genes of cultivated and wild soybeans with other legume species provides useful information that can be used to harness the agronomic, ecological and economic benefits of the soybean crop.

## Figures and Tables

**Figure 1 antioxidants-11-01626-f001:**
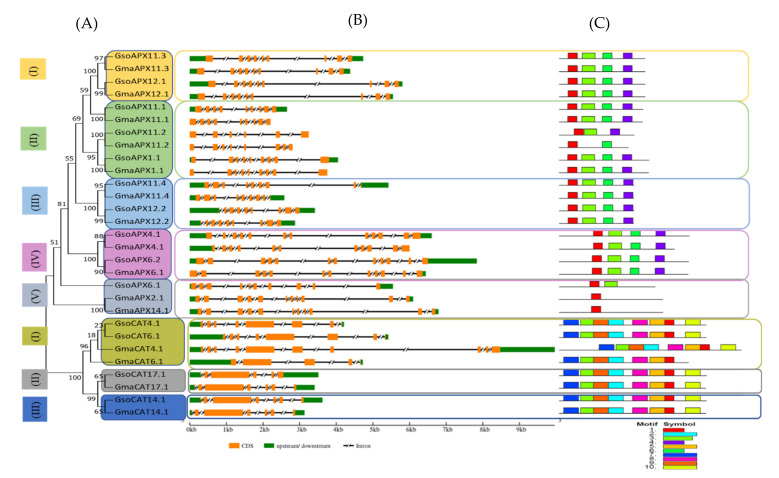
Phylogenetic relationship, exon–intron distribution/gene structure and conserved motifs in *APX* and *CAT* gene families in *G. max* and *G. soja*. (**A**) The phylogenetic tree of the *APX* gene family classified in to five clades highlighted with different colored boxes. (**B**) The untranslated region (UTR), with intron and exon distribution represented by a green box, black scored line and orange boxes, respectively. (**C**) Identification of the conserved motifs in *APX* and *CAT* genes. Each motif is presented by a particular color.

**Figure 2 antioxidants-11-01626-f002:**
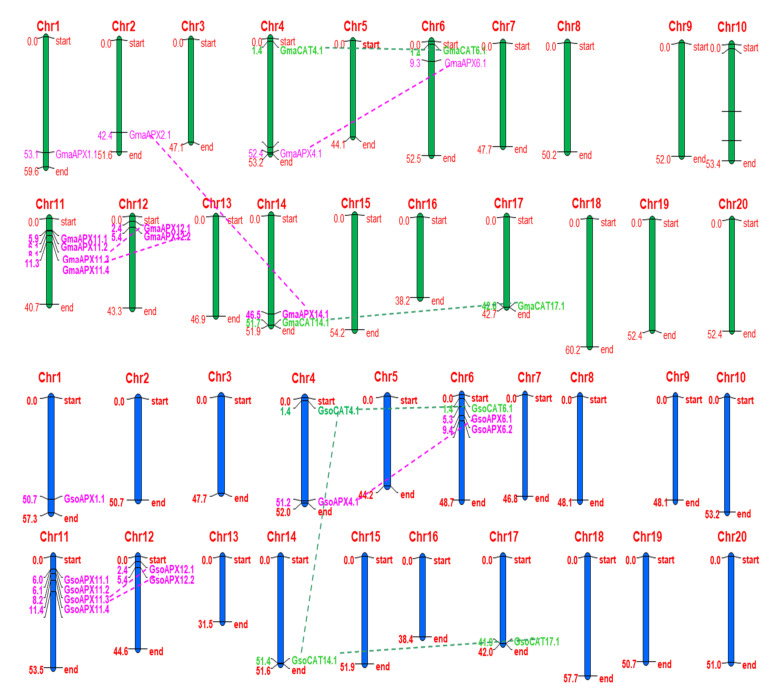
Chromosomal locations and gene duplication events of *APX* and *CAT* genes of *G. max* (green chromosomes) and *G. soja* (blue chromosomes). The scale on the left side of the chromosome is the position of the genes in megabases, and on the right side of each chromosome, gene names correspond to the approximate locations of each *APX* and *CAT* gene. Furthermore, the segmentally duplicated genes are connected by dashed lines represented by the same color as *APX* and *CAT* genes.

**Figure 3 antioxidants-11-01626-f003:**
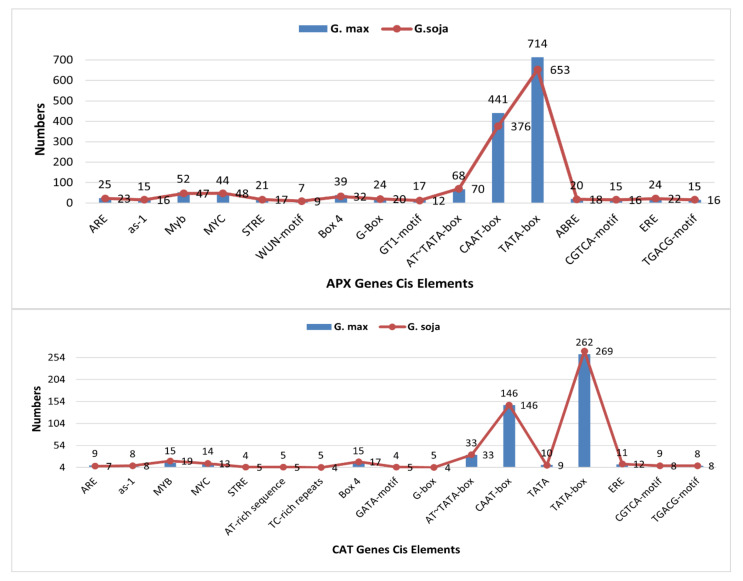
An overview of cis regulatory elements in APX and CAT gene family members of *G. max* and *G. soja*. The blue bars represent the cis-element values for cultivated soybean (*G. max*), whereas the red points within the red line represent the cis-element values for wild soybean (*G. soja*).

**Figure 4 antioxidants-11-01626-f004:**
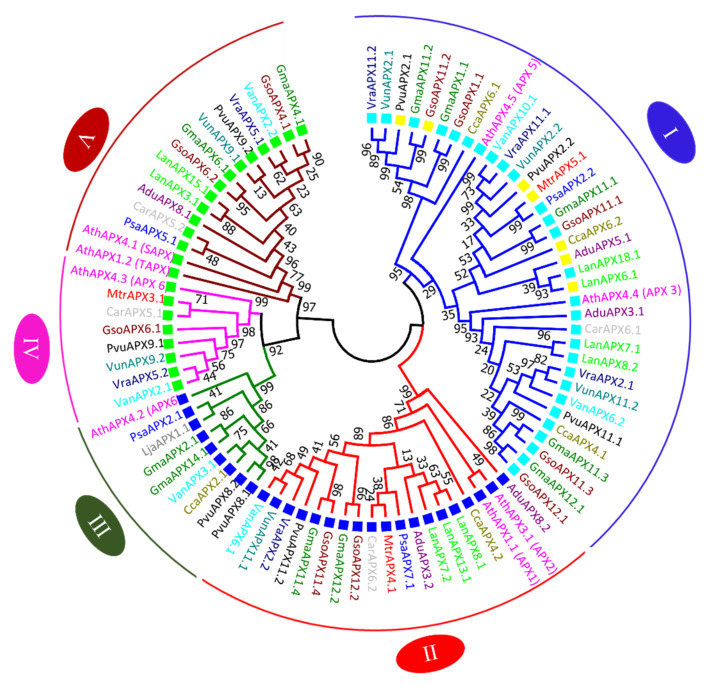
Phylogenic tree of APX (A) and CAT (B) proteins. The tree was constructed in MEGA 7 using the neighbor-joining method with 1000 bootstraps. Different clades of trees are marked with different colors of branch lines: Clade I (blue), Clade II (red), Clade III (green), Clade IV (pink) and Clade V (maroon). The square boxes represent cellular localization: peroxisome (aqua), cytoplasm (blue ), chloroplast (green) and mitochondria (yellow).

**Figure 5 antioxidants-11-01626-f005:**
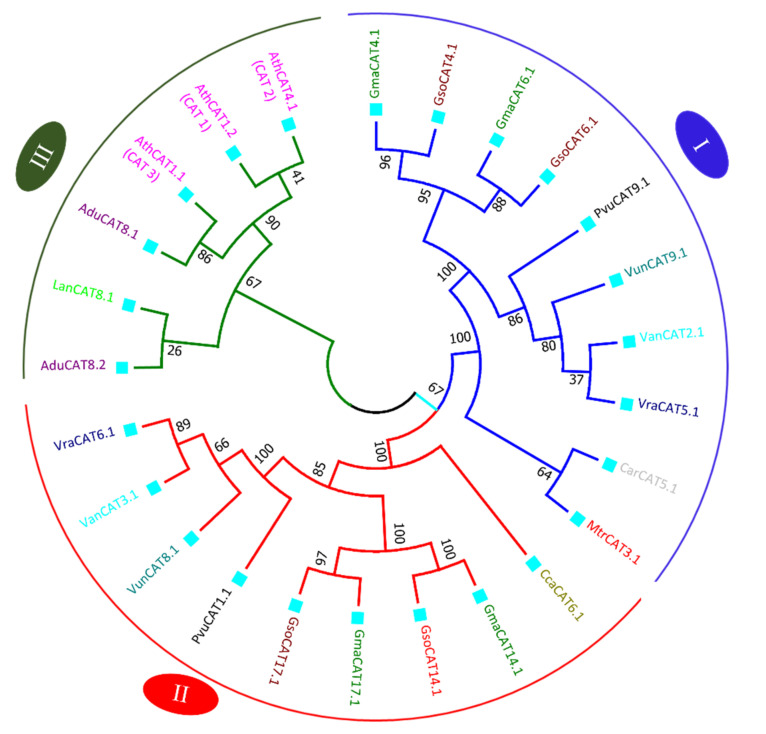
Phylogenic tree of CAT proteins. The tree was constructed in MEGA 7 using the neighbor-joining method with 1000 bootstraps. Different clades of trees are marked with different colors of branch lines: Clade I (blue), Clade II (red) and Clade III (green). The square boxes represent cellular localization: peroxisome (aqua).

**Figure 6 antioxidants-11-01626-f006:**
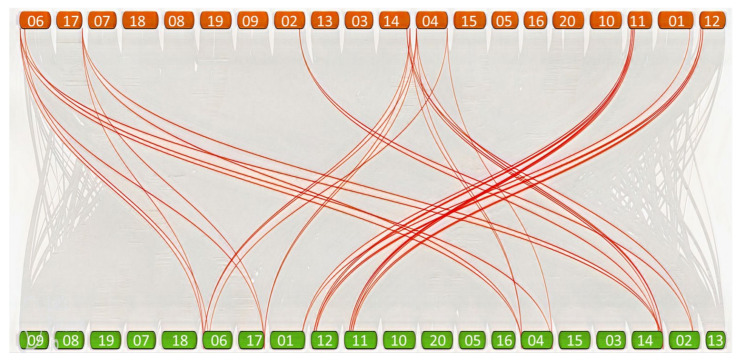
Comparison of *APX* and *CAT* genes between *G. max* and *G. soja* genomes. A genome scale dual-synteny plot between *G. max* (green boxes on the bottom side) and *G. soja* (orange boxes on the upper side) genomes and chromosome number in green and orange boxes, respectively, with *APX* and *CAT* genes represented by red lines.

**Figure 7 antioxidants-11-01626-f007:**
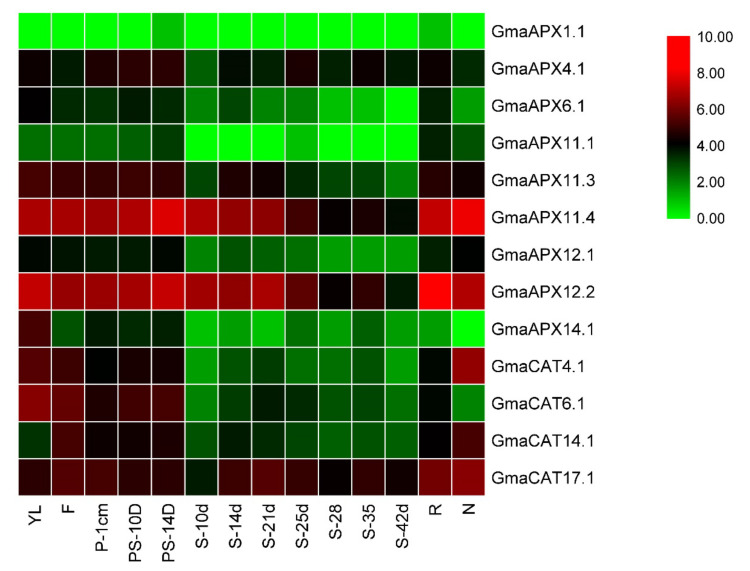
Hierarchical clustering of expression profiles of soybean *APX* and *CAT* genes in different tissues. YL (young leaves), F (flower), P.1cm (one cm pod), PS.10d (10 DAF pod shell), PS.14d (14 DAF pod shell), S.10d (10 DAF seed), S.14d (14 DAF seed), S.21d (21 DAF seed), S.25d (25 DAF seed), S.28d (28 DAF seed), S.35d (35 DAF seed), S.42d (42 DAF seed), R (root), N (nodule). These RNA-seq data were previously generated and deposited in the SoyBase by Shen et al. [56]; the experimental conditions used to generate these RNA-seq data are presented in detail in [56].

**Figure 8 antioxidants-11-01626-f008:**
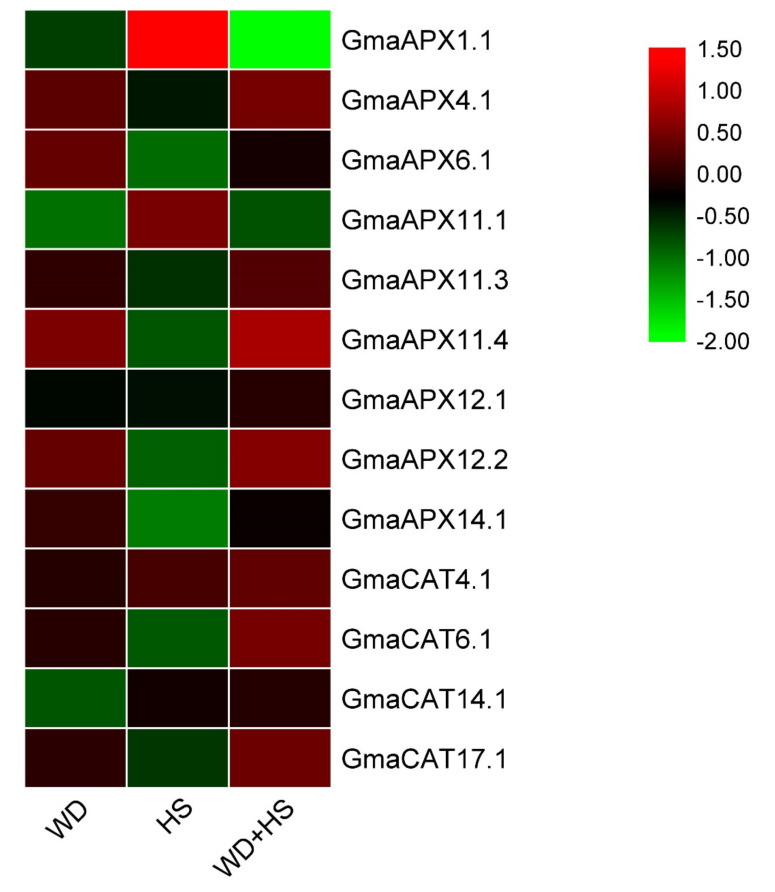
Differential expression of soybean *APX* and *CAT* genes under water deficit (WD), heat stress (HS) and combined water deficit and heat stress (WD + HS). These RNA-seq data were previously generated and deposited in the SoyBase database by Shen et al. [56]; the experimental conditions used to generate these RNA-seq data are presented in detail in [56].

**Figure 9 antioxidants-11-01626-f009:**
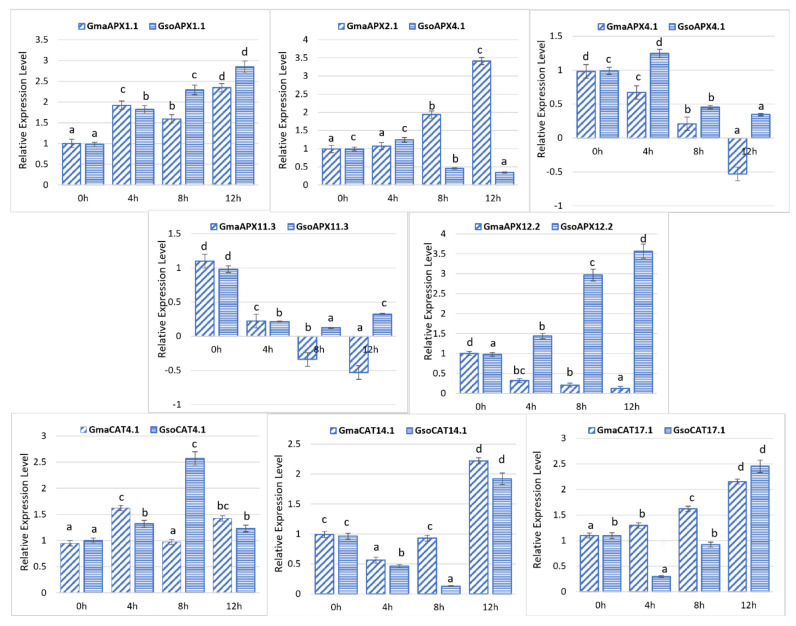
Expression pattern of selected APX and CAT genes of *G. soja* and *G.max* under drought stress. Data depict mean and standard deviation of three replicates (n = 3). Data with the same letters in lowercase (a, b, c and d) above bars indicate no significant differences at the 0.05 level at different time intervals in the soybean genotype according to Duncan’s multiple range test.

**Table 1 antioxidants-11-01626-t001:** Identified APX and CAT genes of wild and cultivated soybeans, including their soybase ID, rename ID, protein domain family and its description, chromosome number, location, orientation, protein size, molecular weight, PI and localization of the genes.

Soybase ID	Rename ID	Protein Domain Family (PFAM)	Chrom. No.	Gene Starting Position	Gene End Position	Orientation	Protein Length	MW (Kda)	Theor. Pi	Localization Cello	Localization Wegoloc
*G. max*											
glyma.Zh13.gnm1.ann1.SoyZH13_01G148100	GmaAPX1.1	PF00041	1	53076021	53079776	+	300	33.81	9.05	Nuclear	Peroxisome
glyma.Zh13.gnm1.ann1.SoyZH13_02G190600	GmaAPX2.1	PF00041	2	42405706	42411803	-	347	37.75	6.24	Extracellular	Chloroplast
glyma.Zh13.gnm1.ann1.SoyZH13_04G223500	GmaAPX4.1	PF00041	4	52362952	52368952	-	386	41.95	6.73	Chloroplast	Chloroplast
glyma.Zh13.gnm1.ann1.SoyZH13_06G109600	GmaAPX6.1	PF00041	6	9291175	9297616	+	432	46.98	7.13	Chloroplast	Chloroplast
glyma.Zh13.gnm1.ann1.SoyZH13_11G076100	GmaAPX11.1	PF00041	11	5894217	5896422	-	280	31.18	9.08	Cytoplasmic	Peroxisome
glyma.Zh13.gnm1.ann1.SoyZH13_11G078900	GmaAPX11.2	PF00041	11	6066134	6068941	+	232	26.64	8.74	Cytoplasmic	Peroxisome
glyma.Zh13.gnm1.ann1.SoyZH13_11G104300	GmaAPX11.3	PF00041	11	8138680	8143056	-	287	31.72	6.62	Cytoplasmic	Peroxisome
glyma.Zh13.gnm1.ann1.SoyZH13_11G142000	GmaAPX11.4	PF00041	11	11329025	11331607	+	250	27.05	5.5	Cytoplasmic	Cytoplasm
glyma.Zh13.gnm1.ann1.SoyZH13_12G031100	GmaAPX12.1	PF00041	12	2425213	2430761	-	287	31.76	6.27	Cytoplasmic	Peroxisome
glyma.Zh13.gnm1.ann1.SoyZH13_12G067000	GmaAPX12.2	PF00041	12	5412074	5414947	-	250	27.13	5.65	Cytoplasmic	Cytoplasm
glyma.Zh13.gnm1.ann1.SoyZH13_14G162900	GmaAPX14.1	PF00041	14	46547958	46554747	-	347	37.92	6.76	Extracellular	Chloroplast
glyma.Zh13.gnm1.ann1.SoyZH13_04G016700	GmaCAT4.1	PF00199	4	1374434	1384397	-	611	69.88	6.61	Peroxisomal	Peroxisome
glyma.Zh13.gnm1.ann1.SoyZH13_06G016600	GmaCAT6.1	PF00199	6	1350172	1354895	-	434	49.79	6.26	Peroxisomal	Peroxisome
glyma.Zh13.gnm1.ann1.SoyZH13_14G205900	GmaCAT14.1	PF00199	14	51701327	51704461	+	492	56.91	6.77	Peroxisomal	Peroxisome
glyma.Zh13.gnm1.ann1.SoyZH13_17G248900	GmaCAT17.1	PF00199	17	42607546	42610955	+	492	56.95	6.77	Peroxisomal	Peroxisome
***G. soja***											
glyso.W05.gnm1.ann1.Glysoja.01G001741	GsoAPX1.1	PF00041	1	50740356	50744403	+	300	33.8	9.05	Nuclear	Peroxisome
glyso.W05.gnm1.ann1.Glysoja.04G010747	GsoAPX4.1	PF00041	4	51176378	51182983	-	435	47.3	7.1	Chloroplast	Chloroplast
glyso.W05.gnm1.ann1.Glysoja.06G013959	GsoAPX6.1	PF00041	6	5275075	5280621	+	321	34.4	7.56	Chloroplast	Cytoplasm
glyso.W05.gnm1.ann1.Glysoja.06G014412	GsoAPX6.2	PF00041	6	9350953	9358789	+	432	47.1	7.13	Chloroplast	Chloroplast
glyso.W05.gnm1.ann1.Glysoja.11G029301	GsoAPX11.1	PF00041	11	5950676	5953331	-	280	31.2	9.08	Cytoplasmic	Peroxisome
glyso.W05.gnm1.ann1.Glysoja.11G029327	GsoAPX11.2	PF00041	11	6122599	6125845	+	250	28.1	8.65	Mitochondrial	Plasma Membrane
glyso.W05.gnm1.ann1.Glysoja.11G029583	GsoAPX11.3	PF00041	11	8214021	8218754	-	287	31.7	6.62	Cytoplasmic	Peroxisome
glyso.W05.gnm1.ann1.Glysoja.11G029970	GsoAPX11.4	PF00041	11	11420791	11423997	+	250	27.1	5.51	Cytoplasmic	Cytoplasm
glyso.W05.gnm1.ann1.Glysoja.12G033246	GsoAPX12.1	PF00041	12	2440656	2446460	-	287	31.8	6.27	Cytoplasmic	Peroxisome
glyso.W05.gnm1.ann1.Glysoja.12G033638	GsoAPX12.2	PF00041	12	5431487	5434903	-	250	27.1	5.52	Cytoplasmic	Cytoplasm
glyso.W05.gnm1.ann1.Glysoja.04G008496	GsoCAT4.1	PF00199	4	1367980	1372190	-	492	56.7	6.8	Peroxisomal	Peroxisome
glyso.W05.gnm1.ann1.Glysoja.06G013473	GsoCAT6.1	PF00199	6	1361430	1366850	-	492	56.7	6.77	Peroxisomal	Peroxisome
glyso.W05.gnm1.ann1.Glysoja.14G039529	GsoCAT14.1	PF00199	14	51391433	51395055	+	492	56.9	6.77	Peroxisomal	Peroxisome
glyso.W05.gnm1.ann1.Glysoja.17G047079	GsoCAT17.1	PF00199	17	41904865	41908375	+	494	57.4	6.93	Peroxisomal	Peroxisome

**Table 2 antioxidants-11-01626-t002:** Gene duplication analysis for the APX and CAT gene families of wild and cultivated soybeans.

Duplicated Pair	Ka	Ks	Ka/Ks	Data (Mya)	Duplicate Type	Purifying Selection	Type
*GmaAPX14.1/GmaAPX2.1*	0.04	0.06	0.69	4.92	Segmental	Yes	Negative or purifying selection
*GmaAPX4.1/GmaAPX6.1*	0.04	0.07	0.56	5.74	Segmental	Yes	Negative or purifying selection
*GmaAPX11.3/GmaAPX12.1*	0.04	0.02	2.26	1.64	Segmental	No	Positive selection
*GmaAPX11.4/GmaAPX12.2*	0.05	0.05	1.11	4.10	Segmental	No	Positive selection
*GsoAPX4.1/GsoAPX6.2*	0.03	0.09	0.36	7.38	Segmental	Yes	Negative or purifying selection
*GsoAPX11.4/GsoAPX12.2*	0.04	0.09	0.39	7.38	Segmental	Yes	Negative or purifying selection
*GsoAPX11.3/GsoAPX12.1*	0.02	0.09	0.23	7.38	Segmental	Yes	Negative or purifying selection
*GmaCAT4.1/GmaCAT6.1*	0.03	0.1	0.3	8.20	Segmental	Yes	Negative or purifying selection
*GmaCAT14.1/GmaCAT17.1*	0.01	0.13	0.05	10.66	Segmental	Yes	Negative or purifying selection
*GsoCAT6.1/GsoCAT4.1*	0.01	0.08	0.08	6.56	Segmental	Yes	Negative or purifying selection
*GsoCAT4.1/GsoCAT14.1*	0.06	0.74	0.08	60.66	Segmental	Yes	Negative or purifying selection
*GsoCAT14.1/GsoCAT17.1*	0.02	0.14	0.14	11.48	Segmental	Yes	Negative or purifying selection

## Data Availability

Data is contained within the article and Appendix A.

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
