# Peer review of "Whole-Genome Identification of APX and CAT Gene Families in Cultivated and Wild Soybeans and Their Regulatory Function in Plant Development and Stress Response"

_antioxidants, 2022, doi:10.3390/antiox11081626_

Round 1

Reviewer 1 Report

I am strongly satisfied from the new modyfied version of the manuscript. The authors suplemented it with wet-lab validation of the in silico obtained results, which strongly support thier data. They also answer all my question and doubts about all the analysis and choosen examples.

I have no mo issues which could be adressed to this paper.

L686 should be supporting

Author Response

Thanks for the positive response. 

You have only comment i.e., Line 686 should be supporting. 

Author Response: We have corrected it in the revised manuscript. It is supporting. Thanks.

Reviewer 2 Report

In their manuscript Aleem et al. analyze and compare the organization of the ascorbate peroxidases (APXs) and catalases (CATs) genes in wild and domesticated soybeans. They then compare the expression profiles of APX and CAT genes in soybean plants submitted to drought stress using publicly available datasets and carried out qRT-PCR for a subset of these genes under drought stress.  Significant differential response of the respective homologous genes of cultivated and wild soybeans under the drought treatments is an interesting result of this study.

I would find the work more interesting if the proteins subcellular localizations were more precisely referenced. For example in Table 1 the authors used 2 softwares to predict soybean APXs localization with sometimes different output. As Arabidopsis APX proteins localizations are well known (refer for example to http://www.chlorokb.fr for a synthesis and references), the authors may add a column indicating for each soybean gene the closest Arabidopsis homolog(s) and its subcellular localization.

Figure 4: I do not understand the color legend for the localization. What is yellow coding ? what is Aqua ?

I found the nomenclature of A.thaliana confusing. I understand that the authors use the same nomenclature for the genes of all the species studied as they explain line 240. However, Arabidopsis APXs have precise names previously used in the literature that should be clearly exposed in e.g. Fig. 4. If the authors prefer maintaining their nomenclature I would advise indicating in the figure legend that AtAPX4.4 for example corresponds to APX3 (with a reference to Table S1) and idem for the other AtAPX. Please note however, that literature indicates a peroxisomal localization for this protein (experimental localization). The color coding for APX4.4 in figure 4 is cyan (mitochondria ?). Please check.

The authors might also add a color code for thylakoidal peroxidase.

In many instances the sentences are difficult to understand and punctuation is strange.

Line 559: Arabidopsis in not really a crop.

Author Response

Please see the attachment, Authors response to reviewers comments.

Reviewer 3 Report

In this study, they used information from databases and the latest bioinformatics softwares to identify and characterize APX genes in soybeans that had not been fully unveiled. The information obtained from this paper will be very useful when discussing how soybeans cope with oxidative stress during environmental stress and differences in tolerance to environmental stress among cultivars. From these points of view, I believe that this study generally meets the requirements for publication in this journal. However, there are a few points that need to be corrected and explained before publication.

1. In Figure 6 and 8, What do values attached to the color bar represent?

2.  Compared to the data in Fig. 6, the response of APXs and CATs genes to WD and HS is very weak. From this data, it is questionable whether we can really say that APXs and CATs are responding to stress. Could you please explain this point?

3. L662-666, I did not know why the authors could conclude this: the differences in individual gene expression in Fig. 8 are not the same, with some higher in wild species and some higher in cultivated species.

4. Figure8 is difficult to see, please correct it.

Author Response

(The authors gave the same response as above.)
